# Interpersonal discrimination and depressive symptoms among older Black and African American adults

Tomorrow D. Arnold[ORCID][1¤]*, Courtney A. Polenick[1,2]◉, Donovan T. Maust[1,3]◉, Frederic C. Blow[1,3]◉

**1** Department of Psychiatry, University of Michigan, Ann Arbor, MI, United States of America, **2** Institute for Social Research, University of Michigan, Ann Arbor, MI, United States of America, **3** VA Center for Clinical Management Research, VA Ann Arbor Healthcare System, Ann Arbor, MI, United States of America

◉ These authors contributed equally to this work.
¤ Current address: Department of Psychology at University of Tennessee at Chattanooga in Chattanooga, Chattanooga, TN, United States of America
* tomorrow-arnold@utc.edu

**Data Availability Statement:** Data have been attached as a supplemental file.

**Funding:** This work was supported by the University of Michigan Depression Center,

## Abstract

To examine the association between recent experiences of discrimination and depressive symptom presentation and severity among a U.S. sample of older Black and African American adults. A cross-sectional survey of 124 Black and African American adults aged 50 and older in the United States was conducted assessing interpersonal discrimination and depressive symptoms. The Perceived Ethnic Discrimination Questionnaire assessed four forms of interpersonal discrimination. A measure of heightened vigilance to bias assessed anticipatory coping with discrimination experiences. Past-month affective and somatic symptoms of depression were assessed using the Depressive and Somatic Symptoms Scale. All forms of interpersonal racial discrimination were positively associated with greater affective symptom severity. Being avoided, devalued, and threatened or actively physically harmed were associated with greater somatic symptom severity. Vigilant coping was positively associated with affective symptom severity but not somatic symptom severity. Racial discrimination is linked to depression severity among older Black and African American and varies by symptom. This study helps inform work on processes linking discrimination with poorer psychological outcomes and will allow for more effective interventions and prevention efforts that are tailored to older minority populations.

## Introduction

In the United States, about a quarter of the U.S. population will be 65 or older by 2030, with those aged 85 and older tripling in number by 2060 [1]. This trend is seen across racial and ethnic groups. The number of African Americans aged 65 and older is expected to triple by 2050, with Blacks and African Americans expected to account for 18% and 25% of adults aged 65 and older in 2030 and 2060, respectively [1, 2]. Among Blacks and African Americans aged 65 and older, approximately 12% had at least one mental health disorder in the past year [3]. The

Strategic Translational Research Award to Dr. T. Arnold. Dr. T. Arnold was supported by the National Institute of Mental Health [grant number T32 MH073553-11]. Dr. C. Polenick was supported by the National Institute on Aging [grant number K01AG059829]. Dr. D. Maust was supported by the National Institute on Drug Abuse [grant number R01DA045705]. This study was also supported by the National Institutes of Health [grant number P30 AG015281], UMHealthResearch.org supported by the National Center for Advancing Translational Sciences [grant number UL1TR002240], and the Michigan Center for Urban African American Aging Research. The funders had no role in study design, data collection and analysis, decision to publish, or preparation of the manuscript.

**Competing interests:** The authors have declared that no competing interests exist.

increase in the aging population will mean growing numbers of older adults with mental health disorders, which will affect many segments of society including the mental health services.

The minority stress model suggests that those in stigmatized or marginalized social positions experience unique stressors as reviewed by [4]. Racial minoritization may be a source of both acute and chronic stress; these stressors accrue over the lifespan and accumulate, increasing risk of mental health among minoritized populations [5, 6]. Older Blacks and African Americans experience more psychological distress than their White counterparts largely due to chronic stressors such as higher levels of daily stress, racial discrimination, and poverty [7]. However, there is limited research addressing the relationships between coping with discrimination, comorbid mental health disorders, and substance use among African American and Black adults in later life.

One contributing factor to this lack of research is the continued under-recognition and treatment of depression among older people of color [8]. Furthermore, data examining co-occurring mental health among older Black adults is especially limited, with most epidemiologic studies focusing on older adults as a homogeneous group and typically only examining race and ethnicity as a covariate. Intersectionality theory provides a theoretical framework to understand how social identities, including race and gender, shape minoritized groups experiences including in experiences of ageism and racism on mental health outcomes [9]. The intersecting identities of African American and Black older adults result in added challenges due to being associated with multiple marginalized and minoritized groups (i.e., older adults and persons of color) that are both often mis- or under-diagnosed and undertreated [9].

Older Black and African American adults have had a long history of interpersonal experiences of discrimination that likely contribute to both the onset and maintenance of mental health disorders [5, 6]. As older African American and Black populations continue to grow, understanding the roots of racial disparities and ageist attitudes in mental health disorders and treatment will help to refine diagnosis and enhance prevention efforts, treatment practices, and long-term outcomes. The purpose of the present paper is to clarify how interpersonal experiences of discrimination are associated with depression and vigilant coping strategy among older Black and African American adults.

Depressive disorders (i.e., major and persistent depressive disorders) are prevalent and marked by continual sadness or emptiness, feelings of guilt or worthlessness, low energy, hyper- or hyposomnia, loss of interest or pleasure in favored activities (anhedonia), appetite changes, and undetermined aches or pains [10]. Older adults may have lower rates of recognition of depressive symptoms, misdiagnosis of disorders, or outright dismissal of symptoms by clinicians relative to younger adults (e.g., increased cognitive impairment) [11–13]. This is further complicated by race.

Findings regarding the prevalence of depressive disorders and symptoms among older Black adults are mixed. Some studies have reported greater prevalence of major depressive disorder and minor depression among older White than older Black adults, for example, whereas others have found no racial/ethnic differences [12, 14]. Others have found that older Black populations may experience more severe symptoms and more recurrent depressive episodes than Whites [15]. These variations may be due in part to differential manifestation of symptoms among older Black adults, such as more somatic or physical symptoms (e.g., psychomotor symptoms) and some affective or emotion-specific symptoms (i.e., greater anhedonia) but less presentation of neurovegetative symptoms such as sleep problems compared to their White counterparts [12, 16, 17].

Racial minority status may be a source of both acute and chronic stress. Stressors such as threats of racial discrimination, internalized racism, and stereotype threat are often associated

with ethnic minority status [18]. Older Black and African American adults experience more psychological distress than their White counterparts largely due to higher levels of overall daily stress, racism, and discrimination [7, 19]. Racial discrimination has been linked to both poorer physical (e.g., hypertension; cardiovascular complications) and mental health outcomes (e.g., depressive symptoms) as well as an increased risk for all-cause mortality for older Black adults [5, 14, 19–22].

Furthermore, there is evidence that African Americans and Afro-Caribbeans who report higher levels of discrimination are more likely to meet diagnostic criteria for major depressive disorder [23, 24]. However, it remains unclear how discrimination in social interactions impacts depression in older Black and African American adults.

Jones [25] suggests that there are three levels at which racism and racial discrimination occur: institutionalized, interpersonal (or personally mediated), and internalized. Institutionalized racism is structurally built into institutions of practice, law, and customs and manifests in differential access to services, goods, and opportunities by race. Interpersonal discrimination involves making assumptions about the abilities, motivations, and intentions of others due to race as well as acting differently towards someone due to their race. This type of racism includes verbal slurs, suspicion, avoidance (e.g., crossing the street), devaluation (e.g., being surprised at competence), physical threats or harassment, and dehumanization (e.g., police brutality) [25, 26].

Lastly, internalized racism is acceptance by marginalized races of negative notions about one's own abilities and intrinsic worth [25]. This form of racism includes self-devaluation, resignation, or helplessness (e.g., engaging in risky behavior, dropping out of school), and embracing Euro-centric physical features (e.g., using skin lighteners). Understanding how individuals cope with interpersonal discrimination will help identify those who may be most at risk and in need of mental health treatment or support and will inform modifiable interventions targeting social and interpersonal conflict.

Overall, coping refers to behavioral strategies used to manage the impact of stressors [27]. These strategies are often categorized as passive or active depending on whether the individual engages or disengages with the stressor [28]. Certain strategies may be more beneficial than others for offsetting the consequences of discrimination on health [29]. A meta-analysis by Pascoe and Smart Richman found that active coping styles (e.g., confrontation, positive reappraisal) are most helpful in dealing with stress associated with racial discrimination among people of color [5].

By contrast, there is some evidence that using vigilant, or anticipatory, coping strategies to prepare for potential discriminatory interactions (e.g., expecting to be called racial or ethnic slurs or to receive poor service and treatment) are linked to adverse physical (e.g., higher odds of hypertension) [30] and mental health outcomes. For example, Himmelstein and colleagues [31] found that use of vigilant coping strategies (i.e., proactively preparing for the experience of discrimination) mediated the relationship between discrimination and stress, which in turn, increased depression among Black adults. That is, anticipating discrimination by increasing vigilance may exacerbate stress and depressive symptoms [31–33].

Research suggests that experiences of discrimination and chronic vigilance to prejudice and discrimination are linked to increased psychological distress and stress [33]. There is also evidence that vigilant coping contributes to racial differences in the link between stress and health outcomes [32, 34]. However, research is limited on how discrimination impacts individuals with potential comorbid mental health issues in Black and Africans American adults, particularly older populations. The present paper seeks to understand how both perceived discrimination and discrimination-related anticipatory coping style impacts depressive symptoms in late life among Black and African American adults.

Discrimination is associated with poorer mental health outcomes among older adults, particularly depression. Studies suggest that vigilance to discrimination as a coping strategy is also associated with increased depression [5, 32, 33]. Data examining levels of discrimination as well as vigilant coping among older Black and African American populations is sparse, limiting understanding of depression in the face of discrimination among these groups in later life.

The present study had two aims: 1) determine whether different forms of interpersonal racial/ethnic discrimination were associated with depressive symptom severity, and 2) determine whether vigilant coping with discrimination was associated with depressive symptom severity among older Black and African American adults. With those aims in mind, we hypothesized that:

1. Frequency of racial/ethnic interpersonal discrimination would be positively associated with the number of affective and somatic depressive symptoms.

2. Vigilant coping would be positively associated with severity of somatic and affective depressive symptoms.

## Methods

### Participants

Participants were adults aged 50 and older who identified as Black or African American and who were recruited as a part of a study to understand depression and discrimination as risk factors for alcohol, cannabis, and prescription opioid or sedative-tranquilizers use. Participants were recruited between September 20, 2019 and March 22, 2020 from the Midwestern, Mid-Atlantic, and Southern regions of the United States. Participants were recruited through flyers, ads via physical community boards and social media, affiliated research volunteer websites, and the Healthier Black Elders Center (HBEC) Participant Resource Pool, a collaboration between Wayne State University's Institute of Gerontology and University of Michigan's Institute for Social Research (Participant Resource Pool for Minority Health and Aging Research, IRB#: 119102B3E) [35].

Participants directly received a link from the study team or were redirected from social media platforms when expressing interest via an anonymously linked Qualtrics page. This page included detailed information about the purpose of the study, benefits, risk, and compensation ($30) and an electronic version of the waiver of written documentation of consent was provided to be saved and downloaded. To proceed, participants had to choose, 'I agree to participate in the study' which implied consent to continue with the survey. This study was approved by the University of Michigan Medicine Institutional Review Board (Study #: HUM00166210).

### Measures

**Depressive symptoms.** The 22-item Depression and Somatic Symptoms Scale (DSSS) [36, 37] is a self-administered scale that assesses affective or emotion-based symptoms (e.g., depressed mood or tearfulness, inability to concentrate) and somatic or physical symptoms (e.g., soreness in more than half of body, dizziness). The DSSS consists of a 12-item Depression Subscale and 10-item Somatic Subscale; each item assesses symptom severity in the past 30 days on a 4-point scale. Responses include: 'Absent' (no symptoms [0]); 'Mild' (symptom causes slight discomfort or disturbance [1]); 'Moderate' (significant discomfort or disturbance [2]); and 'Severe' (very significant discomfort or disturbance [3]). Higher scores in the Depression Subscale indicate greater affective symptom severity, while higher scores in the Somatic

Subscale indicate greater physical symptom severity. The DSSS and its subscales have good test-retest reliability and internal consistency. The Cronbach's alphas for the present sample were .923, .902, and .820 for the total DSSS (overall symptoms), the Depression Subscale (affective or emotion-based symptoms), and the Somatic Subscale (physical symptoms), respectively.

**Racial/ethnic discrimination.** The 17-item Brief Perceived Ethnic Discrimination Questionnaire was used to assess four domains of discrimination related to race or ethnicity experienced over the past month: verbal rejection, avoidance, devaluation, and threats of violence and aggression [38]. The Verbal Rejection Subscale includes three items assessing frequency of experiencing offensive comments about one's racial/ethnic group and being called derogatory racial or ethnic names. The Avoidance Subscale includes three items assessing frequency that others avoided contact or interaction with them due to their race/ethnicity. The Devaluation Subscale includes six items measuring frequency that others implied or made negative assumptions about them or their racial/ethnic group (e.g., implying that they are unintelligent or dangerous). The Aggression and Threats of Violence Subscale includes five items assessing frequency of being threatened with harm or actively harmed (e.g., having property damaged).

Items measured frequency of experiencing these occurrences in the past month on a 7-point scale, ranging from 'Never' (0) to 'Very Often' (6). The total score was calculated as the mean of all 17 items while subscale scores were calculated based on the average of the corresponding items. Scores ranged from 0 to 6 with higher scores indicating more frequent experiences of racial/ethnic discrimination. For the present sample, Cronbach's alpha for overall racial/ethnic discrimination was .927. Alphas for the subscales included .824 (Verbal Rejection), .933 (Avoidance), .959 (Devaluation), and .777 (Aggression and Threats of Violence).

**Vigilant coping.** The 6-item Heightened Vigilance Scale (HVS) is an unpublished scale developed for the Detroit Area Study [39] to assess vigilance for discrimination which has been used to various studies examining racism in the context of psychological distress, loneliness, and microaggressions [40, 41]. Participants indicated how often they engaged in six behaviors in their day-to-day life in response to discrimination, such as preparing for insults before leaving home and being extra careful about appearance to avoid harassment or poor service. Items measured frequency of doing these behaviors on a 6-point scale with response choices ranging from 'Never' (0) to 'Almost every day' (5). Scores were calculated as the sum of the items and ranged from 0 to 30 with higher scores indicating greater use of vigilance coping strategies for discrimination. The Cronbach's alpha for the current sample was .835.

**Covariates.** We controlled characteristics that may impact racial/ethnic discrimination as well as symptoms of depression. Covariates consisted of age in years; gender (0 = cisgender women, 1 = cisgender man); and self-reported diagnosed psychiatric conditions (i.e., depression, anxiety, or other psychiatric or mental health conditions). A dichotomous variable for psychiatric conditions was created such that 0 represented no psychiatric or mental health conditions and 1 represented at least one condition.

## Procedure

Participants expressing interest were screened on inclusion criteria via an anonymous Qualtrics survey and, if eligible, were redirected to another anonymous Qualtrics page where they consented to participation in the study and an electronic version of the waiver of written documentation of consent was provided. After completion of informed consent, the subject received access to the survey to record their demographic and basic health information as well as relevant measures and questionnaires. Once this survey was completed, each participant was given the option to provide a mailing address to receive a $30 gift card for their time.

## Analytic strategy

Missing responses were minimal (no more than 7.3% on any variable) and were addressed by multiple imputation in IBM SPSS Version 27.0 with the fully conditional specification method, which is used when the pattern of missing data is monotone or non-monotone using 10 imputation datasets [42]. In this model, key analytic variables (i.e., depressive symptoms, interpersonal discrimination, and vigilant coping) were included in adjusting for significant, relevant covariates (i.e., age, gender [0 = cisgender women, 1 = cisgender men], and psychiatric conditions (0 = none, 1 = one or more conditions)) based on preliminary bivariate correlations with participants demographics, health variables, and key analytic variables. This allowed for data to be imputed while specifying select variables as predictors only (i.e., covariates and variables that were not missing data) and/or variables that needed to be imputed.

The aims of the present study were to 1) determine whether different forms of interpersonal racial/ethnic discrimination were associated with depressive symptom severity, and 2) determine whether vigilant coping with discrimination was associated with depressive symptom severity among older Black and African American adults. For the first aim, separate unadjusted and adjusted multiple general linear regression models were conducted to determine whether each of the four forms of interpersonal racial/ethnic discrimination was associated with affective and somatic depressive symptom severity in separate models. This approach was used due to the high collinearity between the PEDQ subscales and differences in presentation of depressive symptoms in previous literature [12, 16, 17]. To address the second aim, additional unadjusted and adjusted general linear regression models was conducted to determine whether vigilant coping was associated with affective and somatic depressive symptom severity. All models adjusted for significant, relevant covariates including age, gender, and psychiatric conditions. All descriptives and analyses were also conducted using SPSS Version 27.

## Results

### Sample descriptives

Table 1 displays summary statistics for sample. The sample consisted of 124 adults aged 50 to 80 years of age ($M$ = 57.19, $SE$ = 0.61); 68.5% were women. About 25.8% were married and the majority reported at least some college-level education (79.9%). Approximately 7.3% of participants were multiracial and 2.4% of the sample was also Latinx. Of the sample, 43.5% reported currently having depression, 33.9% reported having anxiety, and 12.1% reported having other psychiatric conditions.

On average, the sample reported having a total of 14.83 ($SE$ = 0.46) depressive symptoms overall with means of 7.69 ($SE$ = 0.27) affective symptoms and 7.14 ($SE$ = 0.23) somatic symptoms in the past month. In terms of severity, mean total depressive symptom severity was 24.12 ($SE$ = 1.11); mean severities for affective symptoms and somatic symptoms were 14.73 ($SE$ = 0.73) and 9.68 ($SE$ = 0.50), respectively. In terms of interpersonal discrimination, the mean total score on the PEDQ was 1.25 ($SE$ = 0.11), with subscale scores ranging from 0.39 (Threats of violence and physical aggression) to 1.68 (Verbal rejection). The average score for vigilant coping was 16.17 ($SE$ = 0.70).

### Depressive symptom severity

Multiple general linear regression models were used to examine the association between a) types of interpersonal discrimination and symptom severity and b) vigilance to discrimination and symptom severity. Four separate models were conducted to determine the association between individual levels of discrimination and severity of affective and somatic symptoms. Analyses revealed that experiencing verbal rejection ($b$ = 0.82, $p$ = .030), being avoided

**Table 1. Sample characteristics (*N* = 124).**

| | *M* (*SE*) | Frequency (n) |
|---|---|---|
| **Age (range 50–80 years)** | 57.19 (0.61) | |
| **Women (%)** | | 68.5 (85) |
| **Multiracial (%)** | | 7.3 (9) |
| **Latinx (%)** | | 2.4 (3) |
| **Education (%)** | | |
| High school/General education development diploma (GED) | | 20.2 (25) |
| Some college | | 25.0 (31) |
| Associate's/2-year college degree | | 12.9 (16) |
| 4-year college degree | | 21.0 (26) |
| Graduate/professional degree | | 21.0 (26) |
| **Marital status (% married)** | | 25.8 (32) |
| **Employment status (%)** | | |
| Unemployed | | 23.4 (29) |
| Employed (part-time or full-time) | | 54.8 (68) |
| Fully retired | | 21.8 (27) |
| **Chronic health conditions (range 0–8 conditions)** | 2.00 (0.14) | |
| **Diagnosed depression (%)** | | 43.5 (54) |
| **Diagnosed anxiety (%)** | | 33.9 (42) |
| **Other diagnosed psychiatric condition (%)** | | 12.1 (15) |
| **Depressive symptoms** | | |
| Overall severity (range 0–50) | 24.12 (1.11) | |
| Affective symptom severity (range 0–32) | 14.73 (0.73) | |
| Somatic symptom severity (range 0–26) | 9.68 (0.50) | |
| ***Discrimination*** | | |
| **Perceived ethnic discrimination (range 0–6)** | 1.25 (0.11) | |
| Verbal rejection | 1.68 (0.16) | |
| Avoidance | 1.38 (0.15) | |
| Devaluating actions | 1.55 (0.17) | |
| Threats of violence and physical aggression | 0.39 (0.05) | |
| **Vigilant coping (range 0–30)** | 16.17 (0.70) | |

($b$ = 1.45, $p$ < .001), being devalued by others implying or making negative assumptions ($b$ = 0.88, $p$ = .006), being threatened with harm or actively harmed ($b$ = 1.58, $p$ = .014), and vigilance to discrimination ($b$ = 0.27, $p$ < .001) were all associated with greater affective symptom severity (see Table 2). Analyses also revealed that being avoided ($b$ = 0.75, p = .006), being devalued ($b$ = 0.50, $p$ = .039), and being threatened or harmed because of one's race/ethnicity but not verbal rejection were associated with greater somatic symptom severity (see Table 3).

## Discussion

This study demonstrates the associations of distinctive levels of racial and ethnic discrimination as well as vigilance to discrimination with presentation of depressive symptoms among older Black and African American adults, independent of current psychiatric diagnosis. This is one of the few studies to examine specific manifestations of depressive symptoms in relation to discrimination instead of composite scores. The present findings add to the extant literature on the link between 1) discrimination and depression in Black populations [21, 22] and 2) vigilant coping with discrimination and depressive symptoms [31, 32]. Findings suggest that

**Table 2. General linear model for affective depressive symptom severity regressed onto form of discrimination and awareness of discrimination among older African American and Black adults adjusted for covariates (N = 124).**

| Model | Unadjusted Models | | | | | Adjusted Models | | | | |
|---|---|---|---|---|---|---|---|---|---|---|
| | *B* | *SE* | 95% CI | | *p* | *b* | *SE* | 95% CI | | *p* |
| | | | LL | UL | | | | LL | UL | |
| Model 1: Verbal Rejection | 0.81 | 0.46 | -0.09 | 1.70 | .077 | 0.82 | 0.38 | 0.08 | 1.56 | .030 |
| Model 2: Avoidance | 1.36 | 0.44 | 0.491 | 2.224 | .002 | 1.45 | 0.36 | 0.75 | 2.15 | < .001 |
| Model 3: Devaluating actions | 0.78 | 0.40 | 0.002 | 1.55 | .049 | 0.88 | 0.32 | 0.25 | 1.51 | .006 |
| Model 4: Threats of violence and physical aggression | 1.43 | 0.78 | -0.10 | 2.96 | .066 | 1.58 | 0.65 | 0.32 | 2.85 | .014 |
| Vigilance to discrimination | 0.30 | 0.09 | 0.12 | 0.48 | .001 | 0.27 | 0.08 | 0.12 | 0.42 | < .001 |

Note. Adjusted models controlled for age, gender (1 = men, 0 = women), and psychiatric diagnosis (1 = at least one diagnosis, 0 = no diagnosis). CI = confidence interval; LL = lower limit; UL = upper limit.

differences in forms of interpersonal discrimination experienced are distinctly associated with presentation of affective versus somatic symptoms, which has implications for the targeting of interventions and treatments for depressive symptoms and depressive disorders among older Black and African American adults.

In the present study, more frequent experiences of all four levels of discrimination (i.e., verbal rejection, avoidance, devaluation, and threats of violence and aggression) were associated with more severe affective depressive symptoms. These findings are consistent with our hypothesis and previous research showing that discrimination and reactions to racial discrimination include increased negative affect [5] and poorer mental health outcomes in marginalized groups, including Black and African American adults (e.g., increased psychological distress, global scores of depression) [22, 43, 44]. This may be particularly true for recently experienced discrimination relative to lifetime discrimination [5]. Racial discrimination is an uncontrollable stressor and research has found that lack of control over stressors is associated with symptoms of depression [45]. Furthermore, there is evidence that interpersonal forms of racism (e.g., exclusion) are associated with greater negative affect. However, most research has focused on younger populations. For example, Stock and colleagues [46] used a social exclusion paradigm in a sample of Black and African American young adults. They found that racially based social exclusion predicted lower perceived control and greater negative affect, particularly among those who had experienced more discrimination in the past.

These findings suggest that interpersonal discrimination may be elevated in the presence of depressive symptoms, although the type of symptoms and their severity may vary. Further

**Table 3. General linear model for somatic depressive symptom severity regressed onto form of discrimination and awareness of discrimination among older African American and Black adults adjusted for covariates (N = 124).**

| Model | Unadjusted Models | | | | | Adjusted Models | | | | |
|---|---|---|---|---|---|---|---|---|---|---|
| | *B* | *SE* | 95% CI | | *p* | *b* | *SE* | 95% CI | | *p* |
| | | | LL | UL | | | | LL | UL | |
| Model 1: Verbal Rejection | 0.31 | 0.82 | -0.35 | 0.87 | .411 | 0.26 | 0.26 | -0.29 | 0.80 | .354 |
| Model 2: Avoidance | 0.71 | 0.31 | 0.11 | 1.31 | .021 | 0.75 | 0.27 | 0.22 | 1.29 | .006 |
| Model 3: Devaluating actions | 0.44 | 0.27 | -0.09 | 0.97 | .101 | 0.50 | 0.24 | 0.03 | 0.97 | .039 |
| Model 4: Threats of violence and physical aggression | 1.28 | 0.52 | 0.26 | 2.31 | .014 | 1.38 | 0.47 | 0.46 | 2.29 | .003 |
| Vigilance to discrimination | 0.09 | 0.06 | -0.04 | 0.22 | .167 | 0.07 | 0.06 | -0.04 | 0.19 | .211 |

*Note.* Adjusted models controlled for age, gender (1 = cisgender men, 0 = cisgender women), and psychiatric diagnosis (1 = at least one diagnosis, 0 = no diagnosis). CI = confidence interval; LL = lower limit; UL = upper limit.

exploratory analyses of the current sample revealed that more frequent experiences of being avoided or not interacted with and devalued by others implying or making negative assumptions because of one's race or ethnicity were most strongly related to more severe depressed mood, loss of interest in activities, anhedonia, feelings of anxiousness, decrease in libido, and inability to concentrate. This somewhat mirrors what is known about negative affect in the face of social exclusion.

Excluded individuals experience unpleasant emotions (e.g., sadness and shame) and anticipate exclusion again in the future after experiencing exclusion [46–48]. Those in stigmatized groups (e.g., racial and ethnic minorities, immigrants) are more likely to experience social exclusion chronically [49]. Those experiencing this chronic exclusion, in turn, report experiencing higher levels of depression as well as helplessness and perceived meaninglessness [50]. It is unclear how specific instances of race-based exclusion and devaluation may impact depression in older minority populations. Clinicians are uniquely positioned to address the impact and salience of interpersonal discrimination in those experiencing depression or subclinical symptoms.

Exploratory findings revealed that more frequent experiences of threats of aggression or actively being harmed due to race or ethnicity were most strongly associated with more severe anhedonia, anxiousness, feelings of guilty or being critical of oneself, and suicidal ideation. These findings are consistent with previous research showing that discrimination is associated with increased depressive symptoms [5, 44, 51]. Some previous research has also suggested that older Black adults have increased affective symptoms, particularly anhedonia and sadness compared to White adults [17]. Additional work is needed to understand how these different forms of interpersonal discrimination affect presentation of depressive symptoms and clinical depression in older Black and African American populations.

In partial support of our hypothesis, we also found that greater use of a vigilant coping strategy was associated with more severe affective symptoms. These findings mirror work indicating that vigilant coping may be a risk factor for depression [32]. Hyper-vigilance may contribute to rumination, which involves focusing on the negative emotions, causes, and consequences of a situation or event [41, 52]. This anticipatory style of coping can increase stress-related heightened vigilance (e.g., rumination) related to discrimination and can result in increased depressive symptoms [31, 33].

When considering somatic symptoms, we found that more frequent experiences of avoidance, devaluation, and threats of violence and physical aggression were associated with greater severity of somatic symptoms but were not associated with instances of verbal rejection and vigilant coping. Racial and ethnic discrimination has been positively associated with both depressive symptoms and poor physical health outcomes including hypertension, sleep difficulties, and substance use and related disorders [23, 53]. Consequently, the present findings shed light on forms of discrimination that may have an especially harmful impact on physical aspects of depression and on broader health outcomes among older Black and African American adults who use substances.

Our findings that verbal rejection and vigilant coping were not associated with more severe somatic symptoms suggests that these aspects of discrimination may have a more direct negative impact on emotional than physical aspects of depression in the population. However, more research is needed to clarify these links and better understand underlying mechanisms.

## Conclusions

While this is one of the few studies to examine discrimination, coping, and depression in older Black and African American adults, it is not without limitations. The present study used a

cross-sectional design preventing causal inferences to be made in the links between discrimination and depressive symptoms. Furthermore, the present study did not examine the intersection of social identities, namely age and gender, to determine extent of experiences of discrimination due to ageism, sexism, or other domains of discrimination impacting depression and vigilant coping. Future work should include daily or momentary assessment of depression, vigilant coping, and discrimination such as what has been done in young adult samples [54] to get a clearer picture of the temporal nature of discrimination and attributions of each experience in relation to depressive symptoms and vigilant coping.

While GLMs controlled for psychiatric diagnoses, it should be noted that the present sample had high rates of self-reported diagnosis of depression (43.5%), anxiety (33.9%), and other psychiatric conditions (12.1%). The participants of the present study were a part of a larger study and included only those who use substances and were highly educated (79.9% with some college or higher) with internet access and devices to complete the survey. Thus, these results may not generalize to those who do not engage in substance use, do not have other psychiatric conditions, or have lower education attainment.

Additionally, recall bias may threaten reports of past-month experiences of racial/ethnic discrimination through unintentional minimization, under recognition, or overestimating frequency [55]. Emotional pain from experiencing racism may inhibit recall of such experiences which may result in underreporting [56]. Additional longitudinal research is needed to better understand the temporal relationship of vigilant coping and both affective and somatic symptoms of depression to identify unique risk factors in older Black and African American populations that aid in both clinical assessment and treatment of associated symptoms and disorders.

Despite these limitations, the present study determined that interpersonal discrimination was differently associated with affective versus somatic symptoms of depression severity and indicators in a sample of older Black and African American adults This study suggests that type of interpersonal discrimination may affect some aspects of depression (e.g., verbal rejection and affective symptoms) more than others (e.g., verbal rejection and somatic symptoms). By understanding these processes and how they unfold over time, we will be better equipped to help those manage acute and long-term symptoms of depression in the face of uncontrollable, chronic stressors.

## Supporting information

**S1 File.**
(SAV)

## Author Contributions

**Conceptualization:** Tomorrow D. Arnold.

**Data curation:** Tomorrow D. Arnold.

**Formal analysis:** Tomorrow D. Arnold.

**Funding acquisition:** Tomorrow D. Arnold, Courtney A. Polenick, Frederic C. Blow.

**Investigation:** Tomorrow D. Arnold.

**Methodology:** Tomorrow D. Arnold.

**Project administration:** Tomorrow D. Arnold.

**Resources:** Frederic C. Blow.

**Supervision:** Courtney A. Polenick, Donovan T. Maust, Frederic C. Blow.

**Validation:** Tomorrow D. Arnold.

**Visualization:** Tomorrow D. Arnold.

**Writing – original draft:** Tomorrow D. Arnold.

**Writing – review & editing:** Tomorrow D. Arnold, Courtney A. Polenick, Donovan T. Maust.

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
