## [Decision Letter · Decision Letter 0]

13 Feb 2024

PONE-D-23-27170Interpersonal discrimination and depressive symptoms among older Black and African American adultsPLOS ONE

Dear Dr. Arnold,

Thank you for submitting your manuscript to PLOS ONE. After careful consideration, we feel that it has merit but does not fully meet PLOS ONE’s publication criteria as it currently stands. Therefore, we invite you to submit a revised version of the manuscript that addresses the points raised during the review process.

This manuscript represents a significant contribution to both the academic realm and healthcare services by aiming to determine the relationship between various forms of interpersonal racial/ethnic discrimination and the severity of depressive symptoms. Overall, the work is well-conducted. However, a lack of crucial details that would ensure the robustness of the research has been noted, as highlighted by the reviewers. It is imperative that the suggestions and critiques detailed by them are considered and incorporated, aiming to enhance and enrich the study. Such a process will not only elevate the manuscript's quality but also broaden its impact and relevance in the field of study.

We look forward to receiving your revised manuscript.

Kind regards,

Ricardo de Mattos Russo Rafael, Ph.D.

Academic Editor

PLOS ONE

   "This work was supported by the University of Michigan Depression Center (depressioncenter.org/), Strategic Translational Research Award to TDA. TDA was supported by the National Institute of Mental Health [grant number T32 MH073553-11] (nimh.nih.gov). CAP was supported by the National Institute on Aging [grant number K01AG059829] (nia.nih.gov). DTM was supported by the National Institute on Drug Abuse [grant number R01DA045705] (nida.nih.gov). This study was also supported by the National Institutes of Health [grant number P30 AG015281] (nih.gov), UMHealthResearch.org supported by the National Center for Advancing Translational Sciences [grant number UL1TR002240] (ncats.nih.gov), and the Michigan Center for Urban African American Aging Research (https://mcuaaar.org/)." 

4. In the online submission form, you indicated that "Data is not publicly available due to the study being a pilot. However, data are available upon request."

Reviewers' comments:

Reviewer's Responses to Questions

**Comments to the Author**

1. Is the manuscript technically sound, and do the data support the conclusions?

Reviewer #1: Yes

Reviewer #2: Partly

2. Has the statistical analysis been performed appropriately and rigorously? 

Reviewer #1: Yes

Reviewer #2: I Don't Know

3. Have the authors made all data underlying the findings in their manuscript fully available?

Reviewer #1: Yes

Reviewer #2: No

4. Is the manuscript presented in an intelligible fashion and written in standard English?

Reviewer #1: Yes

Reviewer #2: Yes

5. Review Comments to the Author

Reviewer #1: Title: "Interpersonal Discrimination and Depressive Symptoms among Older Black and African American Adults"

The article aims to investigate the association between recent experiences of discrimination and the presentation and severity of depressive symptoms among a sample of older Black and African American adults in the United States. The objectives include determining whether various forms of interpersonal racial/ethnic discrimination are linked to the severity of depressive symptoms, as well as establishing whether vigilant coping with discrimination is associated with depressive symptom severity in this demographic.

The introduction is proficient in establishing conceptual foundations essential for the discourse and provides a solid basis for the subsequent analyses. The methodology employed is consistent with the study's purpose and executed competently, yielding highly significant findings for the study of racism and its influence on psychological distress. The results and analyses are of great value to the academic field, offering unique and underexplored contributions compared to other studies on this topic.

Particularly noteworthy are the associations identified between vigilance to discrimination and symptom severity. These reveal a positive correlation between vigilant coping and depression symptoms, indicating that a higher level of vigilant coping is linked to more severe affective symptoms of depression, though not with somatic symptom severity. This specific point opens up extensive possibilities for future studies, including those addressing diagnostic and nosographic issues within psychiatry, which must incorporate the production of psychological distress in relation to structural racism into their analytical scope.

This study makes substantial contributions to the field and receives a favorable recommendation for publication.

Reviewer #2: The manuscript approaches a highly important subject although many adjustments are needed. The manuscript aimed to determine whether different forms of interpersonal racial/ethnic discrimination were associated with depressive symptom severity, and determine whether vigilant coping with discrimination was associated with depressive symptom severity among older Black and African American adults. It addresses pertinent issues that could enhance our understanding of interpersonal racial/ethnic discrimination effects. However, it will be necessary to improve important aspects such as format, concepts, methods, results, and references.

Comments

1 - First of all it's important the paper follows the Plos One Submission Guidelines (Especially Reference style, Headings, Page and line numbers).

2 - Introduction - For greater clarity, it would be interesting to eliminate the subsections.

3 - Methods / Participants - “Participants were adults aged 50 and older who identified as Black or African American and who reported they had either used alcohol, cannabis, or prescription opioids or sedative tranquilizers at least once in the past month”. - Why were people selected who reported they had either used alcohol, cannabis, prescription opioids, or sedative tranquilizers at least once in the past month? It would be great to specify.

4 - Methods / Measures / Vigilant coping - Are there other recent studies that used the Heightened Vigilance Scale (HSV)?

5 - Methods / Measures / Covariates - Other covariates were tested in the model? Income? Marital Status? Employment Status?

6 - Methods / Analytic strategy - It would be important to specify the model’s adjustments, residual deviance, and which function was used.

7 - Results - It would be highly important to present results minutely. Please, present the crude and adjusted model. Describing the power of the associations found, not only the p-value.

8 - Conclusions - It would be great to present the conclusions in only one section.

6. PLOS authors have the option to publish the peer review history of their article (what does this mean?). If published, this will include your full peer review and any attached files.

Reviewer #1: **Yes: **Tiago Braga do Espírito Santo

Reviewer #2: No

---

## [Author Response · Author response to Decision Letter 0]

26 Mar 2024

1. Thank you for responding to the earlier critiques. However, there is still some clarifications that would strengthen this paper. The abstract sill includes language that is overly general (e.g., “little is known about patterns of substance use and associated factors among patients with OUD”). Please revise to more accurately represent limitations in this domain, e.g., “But there is still more to learn about how patterns of substance misuse and associated factors among OUD patients vary by gender.”

Thank you for the clarification in feedback. We have removed the generic language in the Background section of the abstract to now read: “There is also still more to learn regarding how factors associated with continued and concurrent use might differ for men and women in methadone maintenance treatment (MMT).”

2. In the Conclusion section of the Abstract indicate why the study results are important.

We thank you for this feedback and have added more detail about why the study results are of importance. 

3. There are two sets of Tables and Figures in the submitted manuscript. Please make sure that only one set is included in the next submission.

We appreciate you pointing out this mistake and have made sure that tables were not submitted twice in this revision.

4. Please add the word patients to the title, e.g., "Gender Differences in Patterns and Correlates of Continued Substance Use among Patients in Methadone Maintenance Treatment"

We appreciate the suggestion and have added “Among Patients” to the title to better reflect the topic.

---

## [Decision Letter · Decision Letter 1]

8 May 2024

Interpersonal discrimination and depressive symptoms among older Black and African American adults

PONE-D-23-27170R1

Dear Dr. Arnold,

We’re pleased to inform you that your manuscript has been judged scientifically suitable for publication and will be formally accepted for publication once it meets all outstanding technical requirements.

Kind regards,

Ricardo de Mattos Russo Rafael, Ph.D.

Academic Editor

PLOS ONE

Reviewers' comments:

Reviewer's Responses to Questions

**Comments to the Author**

1. If the authors have adequately addressed your comments raised in a previous round of review and you feel that this manuscript is now acceptable for publication, you may indicate that here to bypass the “Comments to the Author” section, enter your conflict of interest statement in the “Confidential to Editor” section, and submit your "Accept" recommendation.

Reviewer #1: All comments have been addressed

2. Is the manuscript technically sound, and do the data support the conclusions?

Reviewer #1: Yes

3. Has the statistical analysis been performed appropriately and rigorously? 

Reviewer #1: Yes

4. Have the authors made all data underlying the findings in their manuscript fully available?

Reviewer #1: Yes

5. Is the manuscript presented in an intelligible fashion and written in standard English?

Reviewer #1: Yes

6. Review Comments to the Author

Reviewer #1: The present study presents scientific quality and academic relevance for publication.

The authors responded competently to the comments raised by the reviewers.

the data are collected and the analyzes were carried out competently and carefully.

7. PLOS authors have the option to publish the peer review history of their article (what does this mean?). If published, this will include your full peer review and any attached files.

Reviewer #1: **Yes: **Tiago Braga do Espírito Santo

---

## [Editor Report · Acceptance letter]

13 May 2024

PONE-D-23-27170R1 

PLOS ONE

Dear Dr. Arnold, 

I'm pleased to inform you that your manuscript has been deemed suitable for publication in PLOS ONE. Congratulations! Your manuscript is now being handed over to our production team.

Kind regards, 

on behalf of

Dr. Ricardo de Mattos Russo Rafael 

Academic Editor

PLOS ONE